# Anti-*Leptospira* Antibodies in Buffaloes on Marajó Island

José Diomedes Barbosa [1], Fernanda Monik Silva Martins [1], Eliel Valentim Vieira [1], Ruama Paixão de Lima Silva [1], Henrique dos Anjos Bomjardim [2], Marcos Xavier Silva [3] and Felipe Masiero Salvarani [1,*]

[1] Instituto de Medicina Veterinária, Universidade Federal do Pará, Castanhal 68740-970, PA, Brazil; diomedes@ufpa.br (J.D.B.); nanda.monik08@gmail.com (F.M.S.M.); vieiraeliel2022@gmail.com (E.V.V.); ruama-paixao@hotmail.com (R.P.d.L.S.)

[2] Instituto de Estudos do Trópico Úmido, Universidade Federal do Sul e Sudeste do Pará, Xinguara 68557-335, PA, Brazil; henriquebomjardim@unifesspa.edu.br

[3] Escola de Veterinária (EV), Universidade Federal de Minas Gerais, Belo Horizonte 31270-901, MG, Brazil; vetmarcosxavier@hotmail.com

\* Correspondence: felipems@ufpa.br

**Abstract:** Leptospirosis is a zoonotic disease that has a cosmopolitan geographical distribution, reported in domestic and wild animals, which act as reservoirs and contribute to the spread of microorganisms in the environment. In Brazil, studies on the occurrence of leptospirosis in buffaloes in the Amazon Biome are scarce. The objective of this study was to determine the occurrences of antibodies against *Leptospira* spp., including serovar Hardjo (Bolivia), isolated from cattle in Brazil and not yet tested in buffaloes. A total of 387 blood serum samples of animals from nine municipalities on Marajó Island, State of Pará, northern Brazil, were obtained from a biological sample bank and analyzed using the microscopic agglutination test (MAT). Serology revealed 91.5% (387/354) of the animals tested positive for anti-*Leptospira* antibodies. The presence of various detected serovars may have been related to the local practice of combined rearing of different livestock species, as well as to the contact with wild animals and rodents from adjacent forest areas, all factors that likely facilitated the epidemiological chain of the disease in buffaloes. Among the serovars tested, the serovar Hardjo (Bolivia) was the most prevalent, which was present in 79.3% of the reactive buffaloes. It was important to carry out serological and bacteriological surveys in order to identify the serovars that occurred in the herds, with the objective of designing efficient strategies to control leptospirosis in the production of buffaloes.

**Keywords:** leptospirosis; diagnosis; *Bubalus bubalis*; zoonosis; Amazon Biome

## 1. Introduction

Leptospirosis is an infectious disease caused by the spirochete bacteria of the species *Leptospira*, which includes more than 260 reported serovars [1]. This zoonotic disease has a cosmopolitan geographical distribution, having been reported in domestic and wild animals, which act as reservoirs and contribute to the spread of the bacteria in the environment. The disease is more frequent in hot–humid tropical regions due to climatic characteristics such as abundant rainfall, heat, and humidity, which favor the prevalence of the bacteria in the environment [2,3].

In Brazil, previous studies have identified the occurrences of anti-*Leptospira* antibodies in buffaloes [4–8], but, in the seroepidemiological surveys, none used *Leptospira interrogans* serovar Hardjo (strain Bolivia, genotype Hardjoprajitno), isolated from cattle in Brazil [9,10], so this is the first study to use this strain for the evaluation of MAT anti-*Leptospira* titers in buffaloes.

The primary form of transmission of buffalo leptospirosis is through the contact of susceptible animals with the urine of infected animals through either the skin, mucous membranes, or conjunctiva. In the Amazon region, buffaloes remain in flooded areas for

long periods of time and may develop subclinical infections that favor the epidemiological chain of the disease in a herd [10]. The main clinical signs of leptospirosis that can be observed are abortion and mastitis, which have direct negative impacts on production, causing considerable losses to farmers and the livestock sector [7,11].

The number of buffaloes in Brazil is around 1.4 million heads, and the State of Pará in northern Brazil is responsible for 38.2% of the total number of animals in the country, with Marajó Island accounting for the largest herd [12,13]. However, despite this considerable herd size, the production characteristics in the state, such as extensive grazing alongside other domestic species and deficient hygienic–sanitary control, limit the productive potential of the buffalo sector [14].

Knowledge on the main *Leptospira* serovars circulating in the region is important for adequate management of the disease and to increase productivity, as are the isolation and treatment of diseased animals in addition to preventive vaccinations in herds [10,15,16]

Given the importance of leptospirosis to the buffalo production chain and the possible negative impacts of the disease in the livestock sector, the objective of this study was to determine the occurrence of antibodies against *Leptospira* spp., including *Leptospira interrogans* serovar Hardjo (strain Bolivia) isolated from cattle in Brazil and underused in buffaloes.

## 2. Materials and Methods

The study was conducted using 387 samples of blood serum from Jafarabadi x Murrah crossbred buffaloes (*Bubalus bubalis*) of both sexes, with an average age of 36 months, obtained from the biological sample bank of the Veterinary Hospital of the Institute of Veterinary Medicine of the Federal University of Pará (Instituto de Medicina Veterinária da Universidade Federal do Pará—IMV-UFPA). The samples were collected over 12 months, comprising both the dry and wet seasons in the Amazon region. The sera used in this study were from animals from nine municipalities of Marjó Island, State of Pará: Cachoeira do Arari (n = 60), Santa Cruz do Arari (n = 60), Soure (n = 60), Salvaterra (n = 60), Chaves (n = 60), Melgaço (n = 16), Muaná (n = 3), Ponta de Pedras (n = 60), and Anajás (n = 8).

According to the history of the properties sampled, the buffalo herds were indigenous, reared with low or no sanitary control, only vaccinated for brucellosis and foot-and-mouth disease, and never vaccinated against leptospirosis. The buffaloes were reared in extensive grazing systems alongside other domestic species such as cattle, horses, pigs, and small ruminants. All properties studied had a swath of land with native vegetation with the presence of wild animals, especially rodents, which are possibly reservoirs of the bacteria *Leptospira* spp., and the buffaloes had access to this area. The animals submitted to blood collection had normal health statuses, without reporting from owners of abortion, mastitis, or death near the time of blood collection.

The detection of anti-*Leptospira* spp. antibodies was performed in the Laboratory of Leptospirosis of the Veterinary School of the Federal University of Minas Gerais (Universidade Federal de Minas Gerais) using the microscopic agglutination test, which was considered the gold standard for leptospirosis diagnosis [16,17]. The battery of antigens used for the MAT comprised the serovars Bataviae, Bratislava, Icterohaemorragiae, Pomona, *Leptospira borgpetersenii* serovar Hardjo genotype Hardjobovis, and the *Leptospira interrogans* serovar Hardjo genotype Hardjoprajitno strains CTG, OMS, and Bolivia (Table 1).

**Table 1.** Standards of *Leptospira* spp. per serogroup, serovar, and strain used in the microscopic agglutination test (MAT) to detect anti-*Leptospira* antibodies in buffaloes in Marajó Island, Pará state (Brazilian Amazon).

| Serogroup | Serovar | Strain/Genotype |
|---|---|---|
| Bataviae | Bataviae | Swart |
| Australis | Bratislava | Jez Bratislava |
| Icterohaemorrhagiae | Icterohaemorrhagiae | RGA |
| Pomona | Pomona | Pomona |
| Sejroe | Hardjo | Hardjobovis |
| Sejroe | Hardjo | CTG/Hardjoprajitno |
| Sejroe | Hardjo | OMS/Hardjoprajitno |
| Sejroe | Hardjo | Bolivia/Hardjoprajitno |

The microscopic agglutination test (MAT) was the test used for screening, and animals with titration titers of 1:100 were considered positive. We considered animals as reactive (positive) to the tested serovars if at least 50% agglutination was observed in the respective sera at the 1:100 screening dilution, and non-reactive if less than 50% or no agglutination was observed, according to interpretation parameters by the World Organization for Animal Health [18].

### 3. Results

Of the 387 serum samples tested, 354 tested positive for the presence of anti-*Leptospira* spp. antibodies, indicating an occurrence rate of 91.5% (Table 2).

**Table 2.** Sample size (N) and frequency (%) of reactive and non-reactive animals in the detection of anti-*Leptospira* spp. antibodies in buffaloes from different municipalities in Marajó Island, the State of Pará (Brazilian Amazon), using the microscopic agglutination test (MAT).

| Municipality | Reactive | | Non-Reactive | | Total |
|---|---|---|---|---|---|
| | N | % | N | % | |
| Anajás | 07 | 87.5 | 1 | 12.5 | 8 |
| Chaves | 58 | 96.6 | 2 | 3.4 | 60 |
| Melgaço | 14 | 87.5 | 2 | 12.5 | 16 |
| Muaná | 3 | 100 | 0 | 0 | 3 |
| Ponta de Pedras | 50 | 83.3 | 10 | 16.7 | 60 |
| Salvaterra | 56 | 93.3 | 4 | 6.7 | 60 |
| Santa Cruz do Arari | 50 | 83.3 | 10 | 16.7 | 60 |
| Cachoeira do Arari | 58 | 96.6 | 2 | 3.4 | 60 |
| Soure | 58 | 96.6 | 2 | 3.4 | 60 |
| TOTAL | 354 | 91.5 | 33 | 8.5 | 387 |

Serovar Hardjo (Bolivia) showed the highest reaction percentage, followed by the Hardjo (OMS sample strain) and Hardjo type Hardjobovis serovars (Table 3). Other serovars that reacted in part of the animals analyzed were the Pomona and Hardjo (CTG sample strain). The serovars Bratislava, Icterohaemorrhagiae, and Bataviae reacted less among the buffaloes tested. Of all buffalo tested, 263 (67.2%) were seropositive, up to two serovars in this research.

**Table 3.** Number of animals (N) and frequency of seroactive animals for different serovars of *Leptospira* spp. tested in buffaloes from different municipalities in Marajó Island, State of Pará (Brazilian Amazon), using serum agglutination microscopy (MAT).

| Serovar | N | Frequency (%) |
|---|---|---|
| Bataviae | 9 | 2.3 |
| Bratislava | 47 | 12.1 |
| Hardjo type hardjobovis | 248 | 64.1 |
| Hardjo (CTG sample strain) | 111 | 28.6 |
| Hardjo (OMS sample strain) | 251 | 64.8 |
| Icterohaemorrhagiae | 16 | 4.1 |
| Pomona | 179 | 46.2 |
| Hardjo (Bolivia) | 307 | 79.3 |

## 4. Discussion

We found a higher frequency of reactive buffaloes than those reported in other Brazilian states (37.7% to 43.7% in São Paulo, 27.9% in Paraíba, 70.6% in Maranhão) and previously in Pará, with 80% [4–8]. The high frequency of animals testing positive in all municipalities' studied evidence, the environmental characteristics, and the high incidences in the studied regions were all endemicity factors for leptospirosis [7]. Levels of precipitation were high in the Amazon region, with relative humidity tending to be above 80% and an average annual temperature of 26 °C [19]. These climatic traits favored the prevalence of the bacteria *Leptospira* spp. in the environment and increased the transmission risk of leptospirosis, facilitating the entire epidemiological chain of the disease [20].

In addition to the region's geoclimatic characteristics, the buffalo production system adopted in the Amazon, in which buffaloes are reared extensively alongside other livestock species with no division within and between estates, makes it difficult to adopt hygienic–sanitary management practices, reinforcing the hypothesis that leptospirosis spreads easily among Pará buffalo herds, including on Marajó Island, as previously reported by Barbosa [15]. The high frequency of leptospirosis observed may also be related to the contact of buffaloes with wild animals from adjacent forest areas, namely, rodents and other small mammals, which are possible sources and transmitters of the agent [3].

Buffaloes can also act as key epidemiological agents, serving as important sources and disseminators of leptospirosis due to their immersion behaviors in water [21]. Water is considered a primary route of contamination for new hosts, as it may contain bacteria [22]. As the Amazonian biome is mainly composed of rivers and lakes, it increases the risk of spreading leptospirosis between productive production and wild animals, as well as humans [23]. Given the presence of rodents in the region and the contact of buffaloes with contaminated urine, it is possible that buffaloes are a link in the leptospirosis transmission chain for humans and other ruminants [24].

This is the first study to report leptospiral titers to serovar Hardjo (Bolivia) in buffaloes in Marajó Island, Amazon Biome, Brazil. The high frequency of this serovar in our samples suggests the need to include serovar Hardjo (Bolivia) in the battery of diagnostic antigens for *Leptospira* spp. in buffaloes in Brazil, as animals are traded widely among the Amazon and other regions of Brazil. The high frequencies of other serovars of the Hardjo group in our samples suggest that these serovars, which have a known predilection for cattle, may also have a predilection for buffaloes. The combined rearing of cattle and buffaloes in our study sites may facilitate the interspecific transmission of serovars. Additional serological studies, as well as the isolation and identification of the bacterial agents in buffaloes with the disease, are needed to clarify the occurrence of Hardjo serovars in buffaloes in Brazil [25].

When compared to the data observed by Viana et al.'s [6] study conducted in the State of Pará, we can observe the occurrence of the Serovar Hardjo, whereas they worked with of the serogroup Sejroe, which is considered more common (CTG sample); Hardjo (WHO sample); and Hardjo (Bolivia), which are also from the Sejroe serogroup. With this, we can suggest the need to use different serovars as seroepidemiological research to obtain results

with greater comprehensiveness. When more serovars are included in the surveys [2,20], the diagnosis becomes more specific and is able to identify the most important serovars in the regions studied, optimizing the prophylactic and control measures to be adopted towards the herds.

The prevalence of the Pomona serovar in our samples confirms the infection potential of this serovar in buffalo populations [26]. As buffaloes in Pará are kept in extensive grazing systems in cohabitation with other livestock, the high frequency of the Pomona serovar in buffaloes might be related to cross-infection with pigs [3,5]. The same may be true for the serovar Bratislava, due to contact of buffaloes with horses, one of the hosts and sources of infection by this specific serovar [27].

The serotypes Icterohaemorrhagiae and Bataviae were the least common titers found in the present study among our samples, which can be explained by the extensive breeding system adopted in the farms where buffaloes were bred in greater contact with the sources of infection, like rodents and other wild mammal reservoirs. These animal species are present in higher concentrations near the dwellings of properties and buildings storing feed, which serve as food sources [27–29], although the occurrence of rodents was reported at the farms sampled in this study. The occurrence of animals with antibody titers against the serovar Icterohaemorrhagiae was lower, in addition to other serovars (Bataviae and Bratislava) in the present study and in the literature. However, all serovars of Leptsopira are risk factors for zoonotic occurrence of the disease, reiterating the importance of the use of maximum serovars in the battery of diagnostic tests for leptospirosis in animals and humans. The occurrence of Icterohaemorrhagiae titers in ruminants (in particular in buffaloes) was low. However, as well as other serovars that were reactive in the present study, they still reflected risk factors for the zoonotic occurrence of the disease and its detection, reiterating the importance of the use of different serovars in the antigen batteries of diagnostic tests of leptospirosis [9,10,30].

## 5. Conclusions

Anti-*Leptospira* antibodies were detected in 91.5% (387/354) of buffaloes from nine municipalities of Marajó Island in Pará state in the Amazon region of Brazil. The titers against the serovar Hardjo (strain Bolivia) were the most prevalent, comprising 79.3% of the reactive buffaloes. The presence and prevalence of various serovars detected may have been related to the local practice of combination rearing of different livestock species, as well as with contact with wild animals and rodents from adjacent forest areas, all factors that likely facilitated the epidemiological chain of the disease in buffaloes. It is important to carry out serological and bacteriological surveys in order to identify the serovars that occurred in the herds, with the objective of designing efficient strategies to control leptospirosis in the production of buffaloes.

**Author Contributions:** Conceptualization, J.D.B., M.X.S. and F.M.S.; methodology, F.M.S.M., E.V.V., R.P.d.L.S. and H.d.A.B.; formal analysis, M.X.S. and F.M.S.; investigation, J.D.B., F.M.S.M., E.V.V., R.P.d.L.S. and H.d.A.B.; data curation, J.D.B., M.X.S. and F.M.S.; writing—original draft preparation, J.D.B., M.X.S. and F.M.S.; writing—review and editing, F.M.S.; supervision, J.D.B., M.X.S. and F.M.S.; project administration, J.D.B., M.X.S. and F.M.S. All authors have read and agreed to the published version of the manuscript.

**Funding:** This research received no external funding.

**Institutional Review Board Statement:** The animal study protocol number 8117280421 (ID 001663) was approved by the National Council for Control of Animal Experimentation (CONCEA) and was approved by the Ethic Committee on Animal Use of the Federal University of Para (CEUA/UFPA) in a meeting on 05/27/2022.

**Informed Consent Statement:** Not applicable.

**Data Availability Statement:** Not applicable.

**Acknowledgments:** This article will be published in honor of its author: the veterinarian and Master's degree recipient Mário Arthur da Costa Leal, who died of COVID-19 before he could see his work in the scientific world. The authors are grateful to André Almeida Fernandes, Tatiane Albernaz Ferreira, and Rômulo Cerqueira Leite; CNPq (Conselho Nacional de Desenvolvimento Científico e Tecnológico); FAPESPA (Fundação Amazônia de Amparo a Estudos e Pesquisas do Estado do Pará); CAPES (Coordenação de Aperfeiçoamento de Pessoal de Nível Superior—Finance Code 001); and PROPESP-UFPA (Pró-Reitoria de Pesquisa e Pós-Graduação da Universidade Federal do Pará).

**Conflicts of Interest:** The authors declare no conflict of interest.

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
