# Peer review of "Anti-Leptospira Antibodies in Buffaloes on Marajó Island"

_ruminants, doi:10.3390/ruminants3030017_

Round 1

Reviewer 1 Report

 MDPI (Multidisciplinary Digital Publishing Institute) -Tropical Medicine and Infectious Disease

Manuscript ID -ruminants-2478142

Type-Communication

Title: Anti-Leptospira antibodies in buffaloes in the Marajó Island

Authors:

José Diomedes Barbosa, Fernanda Monik Silva Martins, Eliel Valentim Vieira, Ruama Paixão de Lima Silva, Henrique dos Anjos Bomjardim , Marcos Xavier Silva , Felipe Masiero Salvarani 

comments of reviewer                                        

A brief summary and General concept comments

The objective of this study was to determine the occurrence of MAT anti-Leptospira titres in buffaloes in Brazil, namely titres to serovar Hardjo (strain Bolivia), isolated from cattle in Brazil and not yet tested in buffaloes.

This study can be considered interesting given the recognized importance of buffaloes production in Brazil, the widespread occurrence of Hardjo infections in cattle (the maintenance hosts for serovar Hardjo) in that country, taking also into account that the inclusion of local strains in the antigen batteries can enhance MAT results and that local strain Bolivia was not previously tested in Buffaloes.

The study of leptospiral infection in Buffalos in Brazil is also pertinent considering the vast number of those animals in that country (with Marajó Island accounting for the largest herd), and that they are frequently reared in extensive grazing alongside other domestic animal species, with deficient hygienic-sanitary control. These conditions, together with the geographical and climatic ones, in terms of temperature and humidity, favour the spread of infection, with dire consequences not only to animal production but also to the possibility of spread of infection to humans.

Thus, a deeper knowledge of leptospiral infections in buffaloes in Brazil is useful and important.   

However, the authors should undertake several corrections and improvements in the paper’s text, not only from the point of view of the use of English language, but namely in the report of scientific content.

Things that should be improved

There are several corrections to be done if the article is to be published in a scientific journal. Some corrections are related to minor difficulties in the use of the English language, others require an adjustment on the report of scientific data and ideas and some rephrasing to better express what it’s meant, besides the correction of Leptospira nomenclature to current taxonomy and on the indication of known characterizations of presented strains.      

Thus, the correct distinction between what is a serovar or a strain or a genotype must be evident in both text and tables and in accordance with international nomenclature and previous studies on the characterization of strains. This is important to the right interpretation of results and not to create and contribute to replicate errors.

The origin and reference(s) of the strains should be clear and provided in the paper, namely of strain Bolivia, to which is it given particular importance by the authors. This is a local (isolated in Brazilian Cattle) Hardjo strain, however the indication of its origin and reference is not clearly provided in the paper. Also, along the paper and in Table 1, the presentation of serovars/strains/genotypes is not correctly done, including strain Bolivia, which is presented as belonging to Hardjobovis genotype, when the studies done on its characterization by the authors that isolated readily indicated it belongs to the Hardjoprajitno genotype. 

The discussion should be enriched, namely in the interpretation of such different results in titres obtained for the different Hardjo serovar genotypes, when previous studies referred that the MAT cannot distinguish between Hardjobovis and Hardjoprajitno infections.

Some other specific comments and general suggestions for improvement are further given below.

SPECIFIC COMMENTS OF REVIEWER TO AUTHORS

Line 20

Abstract

Instead of:    “... analyzed using the serum agglutination microscopic test (SAM).

write: “...analyzed using the microscopic agglutination test (MAT).

Lines 34-36

Introduction

Instead of:   “..., none used serovar Hardjo (Bolivia)(incomplete), isolated from cattle in Brazil and underused in buffaloes. Therefore, this is the first one’s studies (wrong English) to evaluate the presence of this serovar in buffaloes (wrong scientific assumption**).

Write:    “..., none used Leptospira interrogans serovar Hardjo (strain Bolivia), isolated from cattle in Brazil (indicate Reference/s*), so this is the first study to use this strain for the evaluation of MAT anti-Leptospira titres in buffaloes.”**

Note 1: According to the literature, the isolation and description of serovar Hardjo strain Bolivia is done in the following papers/references:

*First isolation: Chiareli, O.; Cosate, M.R.V.; Moreira, E.C.; Leite, R.C.; Lobato, F.C.F.; Silva, J.A.; Teixeira, J.F.B.; Marcelino, A.P. 2012. Controle 237 da leptospirose em bovinos de leite com vacina autógena em Santo Antônio do Monte, Minas Gerais. Pesq. Vet.Bras. 2012, 32, 238 633-639.

doi:10.1590/S0100-736X2012000700008

*Further characterization: Cosate, M.R.V., Sakamoto, T., de Oliveira Mendes, T.A. et al. Molecular typing of Leptospira interrogans serovar Hardjo isolates from leptospirosis outbreaks in Brazilian livestock. BMC Vet Res 13, 177 (2017). https://doi.org/10.1186/s12917-017-1081-9

**Note 2: Serology is only serogroup indicative. To evaluate the presence of a given serovar, it is necessary to isolate, identify and characterise the strain involved. Thus, one cannot state that one is evaluating the presence of a given serovar in whatever population if one is only doing a serological study.

Line 40

Introduction

Instead of:    “...and may present subclinical conditions of the disease, which favors the epidemiological.…”

Write:       and may develop subclinical infections, which favor s the epidemiological...

SPECIFIC COMMENTS OF REVIEWER TO AUTHOR (continuation)

Lines 55-56

Introduction

Instead of:  “… including serovar Hardjo (Bolivia) isolated …”

Write: “… including Leptospira interrogans serovar Hardjo (strain Bolivia) isolated…”

Lines 73-74

Materials and Methods

Instead of:    “…with presence wild animals, especially rodents that possibly are reservoirs bacteria Leptospira spp, and....”

Write:       “…with the presence of wild animals, especially rodents, that are possibly are reservoirs of bacteria Leptospira spp, and...”

Lines 80-82

And Table 1

Materials and Methods

Instead of:    “….. The battery of antigens used for the SAM comprised the serovars Bataviae, Bratislava, Hardjo type hardjo-bovis, Hardjo (CTG sample), Hardjo (OMS sample), Icterohaemorragiae, Pomona and Hardjo (Bolivia)...”

Write:       . The battery of antigens used for the SAM MAT comprised the serovars Bataviae, Bratislava, Icterohaemorragiae, Pomona, Leptospira borgpetersenii serovar Hardjo genotype hardjo-bovis Hardjobovis, and Leptospira interrogans serovar Hardjo genotype Hardjoprajitno strains CTG, OMS and Bolivia.”

NOTE 1: Since emphasis in this paper is given on Hardjo titres, it would be best to group together all serovar Hardjo strains in the battery list as well as in Table 1;

NOTE 2- Very important: According to Chiareli et al., Pesq. Vet.Bras. 2012, 32, 238 633-639 and Cosate et al., BMC Vet Res 13, 177 (2017) https://doi.org/10.1186/s12917-017-1081-9, strain Bolivia belongs to genotype Hardjoprajitno and not to genotype Hardjobovis, as it is stated in Table 1. This must be corrected;

Note 3: Hardjobovis is the name of a genotype, not of a serovar. This must be corrected in the Table 1;

Lines 86-87

Table 1

1)      It is not understood why it is stated in the Table title “Data from Chiareli et al. [16]? What does it mean and why is this reference needed? To what is it related? This is not clear and, if possible, the reference could be eliminated from the table or else better explained its relation with the table contents. 

2)      The following corrections are suggested:

 Table 1. Standards of Leptospira spp. per serogroup, serovar and strain used in the microscopic agglutination test (MAT) to detect anti-Leptospira antibodies in buffaloes in Marajó Island, Pará state (Brazilian Amazon).

Serogroup

Serovar

Strain/genotype

Bataviae

Australis

Icterohaemorrhagiae

Pomona

Sejroe

Sejroe

Sejroe

Sejroe

Bataviae

Bratislava

Icterohaemorrhagiae

Pomona

Hardjo

Hardjo

Hardjo

Hardjo

Swart

Jez Bratislava

RGA

Pomona

Hardjobovis

CTG/Hardjoprajitno

OMS/Hardjoprajitno

Bolivia/Hardjoprajitno*

* Chiareli et al., Pesq. Vet.Bras. 2012, 32, 238 633-639; Cosate et al., BMC Vet Res 13, 177 (2017);

SPECIFIC COMMENTS OF REVIEWER TO AUTHOR (continuation-II)

Lines 88-89

Materials and Methods

In:    The microscopic agglutination test (MAT) was the one of screening, considering positive animals with titration (wrong, titration is the procedure to find the titre, not the same than titre) of 1:100.

Consider: In the previous page it was already mentioned that the MAT was used (this is a repetition), but so be it. However, correct as follows: “The microscopic agglutination test (MAT) was the one used of for screening and animals with titration titres of 1:100 were considering positive..

Authors must also make clear if they only screened the sera at this dilution (1:100) ONLY or if they titrated the sera to find the final titre if sera had a titre > 1:100.

Lines 89-92

Materials and Methods

The description authors give for the interpretation of MAT results: “We considered animals with zero result or with one cross as non-reactive to the tested serovars, and animals with two to four crosses as sero-reagent…” is not a correct/ideal way to describe the interpretation of the agglutination. Authors are advised to use the usual descriptions widely used internationally and, that is ++ corresponds to 50% agglutination, thus, should re-write the sentence to something like:

“We considered animals as reactive (positive) to the tested serovars if at least 50% agglutination was observed in the respective sera at the 1:100 screening dilution, and non-reactive if less than 50% or no agglutination was observed.”      

Line 98

Results

Correct: …(Brazilian Amazon) using the microscopic agglutination test (MAT).

Line 100-101

Results

Correct: “…Hardjo (OMS sample strain) and Hardjo type hardjo-bovis Hardjobovis serovars (Table 3). Other serovars 100 that reacted in part of the animals analyzed were the Pomona and Hardjo (CTG sample strain).”

Line 106

Results

Correct: …(Brazilian Amazon) using serum agglutination microscopy (SAM) the microscopic agglutination test (MAT).

Lines 106-107

Table 3

Correct: Where it is written “sample” (CGT and OMS sample) it should be “strain”;

Where is it written “hardjo-bovis” should be “Hardjobovis

Line 108

Discussion

Correct: “…of reactive buffalos than that the one reported in other Brazilian…”

Line 111

Discussion

Instead of:    “…studied, that evidence the environmental characteristics…”

Write: “…studied evidence the environmental characteristics…”

Lines 114-115

Discussion

Correct: “…of the bacteria Leptospira spp..”

Lines 119-120

Discussion

Correct: “…to adopt un hygienic-sanitary management practices,..”

Line 123

Discussion

Instead of:    “…with wild animals, whit rodents, from adjacent forest areas,..”

Write: “…with wild animals, whit namely rodents, from adjacent forest areas,..”

Or, better still: “…with wild animals, namely rodents and other small mammals, from adjacent forest areas...”

Line 129

Discussion

Correct: “…between productive production and wild animals,…”

SPECIFIC COMMENTS OF REVIEWER TO AUTHOR (continuation-III)

Lines 133-134

Discussion

“This is the first study to report the infection of serovar Hardjo (Bolivia)* in buffaloes in Marajó Island, Amazon Biome, Brazil.”

* Note: You cannot state this scientifically because Serology is only serogroup indicative. To evaluate the presence of a given serovar/let alone strain (that is the infection), it is necessary to isolate, identify and characterise the strain involved. To conclude that the buffaloes are infected with Bolivia, you had to isolate the strain.

So, rephrase: “This is the first study to report leptospiral titres to serovar Hardjo (Bolivia) in buffaloes in Marajó Island, Amazon Biome, Brazil.”

Line 141

Discussion

Correct: “…, as well as the isolation and identification of the bacterial agents, (remove comma) in buffaloes…”

Line 143-150

Discussion

This paragraph is confusing.  The Authors are advised to re-write and develop the discussion, explaining better and taking the following into consideration:

1-The results of titres between Hardjo serovars should be similar, because they all belong to the same serovar;

2-Discuss the possible reasons for such different positive titres results between Hardjo CTG and OMS strains (111 and 251) since they are both from Leptospira interrogans serovar Harjo genotype Hardjoprajitno;

3- Discuss the possible reasons for such different positive titres results between Hardjo Leptospira interrogans serovar Hardjo genotype Hardjoprajitno and Leptospira borgpetersenii serovar Harjo genotype Hardjobovis, since previous studies have stated that serological methods are limited in that they can only distinguish the serovars at the serogroup level but cannot differentiate the different genotypes of the Hardjo serovar  (Chideroli et al 2017; Picardeau, 2013). 

In:

Culture Strategies for Isolation of Fastidious Leptospira Serovar Hardjo and Molecular Differentiation of Genotypes Hardjobovis and Hardjoprajitno.

Roberta T. Chideroli, Daniela D. Gonçalves, Suelen A. Suphoronski, Alice F. Alfieri, Amauri A. Alfieri, Admilton G. de Oliveira, Júlio C. de Freitas and Ulisses de Pádua Pereira

Frontiers in Microbiology, November 2017, Vol 8, p 1-8;

https://doi.org/10.3389/fmicb.2017.02155 

Picardeau, M. (2013). Diagnosis and epidemiology of leptospirosis. Med. Mal. Infect. 43, 1–9. doi: 10.1016/j.medmal.2012.11.005

Line 150

Discussion

Correct: “…to be adopted towards the herds.

Line 155-156

Discussion

Correct: “…with horses, one of the hosts and source of infection by this specific serovar.

Note: there are other hosts for Bratislava (namely pigs and hedgehogs)

Line 157-158

Discussion

Correct: “…were the least common titres found in the present study among our samples, which…”

Line 159-160

Discussion

What do you mean?- I doesn’t make sense-Rephrase to a comprehensive sentence: “…and with that the greater contact with the sources of infection with the serovars Icterohaemorrhagiae and Bataviae, like the rat and other animal reservoirs.…”

SPECIFIC COMMENTS OF REVIEWER TO AUTHOR (continuation-IV)

Lines 163-167

Discussion

Correct: “The occurrence of Icterohaemorrhagiae titres in ruminants (in the buffaloes?) was low, it was still, however, as well as other serovars reactive in the present study, they still reflect are risk factors for the zoonotic occurrence of the disease and its detection it is reiterates the importance of the use of different serovars in the antigen battery of diagnostic tests of leptospirosis.

Lines 170-171

Conclusion

Correct: “The titres against serovar Hardjo (strain Bolivia) were the most prevalent.”

Lines 174-175

Conclusion

Correct: “..It is important to carry out serological and bacteriological surveys in order to identify the serovars that occurin the herds...”

The authors should undertake several corrections and improvements in the paper’s text from the point of view of the use of English language, not only related to the linguistic expression itself, but also in the way the scientific content is expressed in English.

Author Response

Reviewer 1

We appreciate all aspects of reviewer 1's comments and suggestions, which made it possible to make this manuscript valid for all readers. Thank you very much for the compliments and empathy in the evaluation.

Line 20

Abstract

Instead of:    “... analyzed using the serum agglutination microscopic test (SAM).…

write: “...analyzed using the microscopic agglutination test (MAT).…

  • Correction performed

Lines 34-36

Introduction

Instead of:   “..., none used serovar Hardjo (Bolivia)(incomplete), isolated from cattle in Brazil and underused in buffaloes. Therefore, this is the first one’s studies (wrong English) to evaluate the presence of this serovar in buffaloes (wrong scientific assumption**).”

Write:    “..., none used Leptospira interrogans serovar Hardjo (strain Bolivia), isolated from cattle in Brazil (indicate Reference/s*), so this is the first study to use this strain for the evaluation of MAT anti-Leptospira titres in buffaloes.”**

Note 1: According to the literature, the isolation and description of serovar Hardjo strain Bolivia is done in the following papers/references:

*First isolation: Chiareli, O.; Cosate, M.R.V.; Moreira, E.C.; Leite, R.C.; Lobato, F.C.F.; Silva, J.A.; Teixeira, J.F.B.; Marcelino, A.P. 2012. Controle 237 da leptospirose em bovinos de leite com vacina autógena em Santo Antônio do Monte, Minas Gerais. Pesq. Vet.Bras. 2012, 32, 238 633-639.

doi:10.1590/S0100-736X2012000700008

*Further characterization: Cosate, M.R.V., Sakamoto, T., de Oliveira Mendes, T.A. et al. Molecular typing of Leptospira interrogans serovar Hardjo isolates from leptospirosis outbreaks in Brazilian livestock. BMC Vet Res 13, 177 (2017). https://doi.org/10.1186/s12917-017-1081-9

  • Correction performed

Line 40

Introduction

Instead of:    “...and may present subclinical conditions of the disease, which favors the epidemiological.…”

Write:       “and may develop subclinical infections, which favor s the epidemiological...”

  • Correction performed

Lines 55-56

Introduction

Instead of:  “… including serovar Hardjo (Bolivia) isolated …”

Write: “… including Leptospira interrogans serovar Hardjo (strain Bolivia) isolated…”

  • Correction performed

ines 73-74

Materials and Methods

Instead of:    “…with presence wild animals, especially rodents that possibly are reservoirs bacteria Leptospira spp, and....”

Write:       “…with the presence of wild animals, especially rodents, that are possibly are reservoirs of bacteria Leptospira spp, and...”

  • Correction performed

Lines 80-82

And Table 1

Materials and Methods

Instead of:    “….. The battery of antigens used for the SAM comprised the serovars Bataviae, Bratislava, Hardjo type hardjo-bovis, Hardjo (CTG sample), Hardjo (OMS sample), Icterohaemorragiae, Pomona and Hardjo (Bolivia)...”

Write:       “. The battery of antigens used for the SAM MAT comprised the serovars Bataviae, Bratislava, Icterohaemorragiae, Pomona, Leptospira borgpetersenii serovar Hardjo genotype hardjo-bovis Hardjobovis, and Leptospira interrogans serovar Hardjo genotype Hardjoprajitno strains CTG, OMS and Bolivia.”

NOTE 1: Since emphasis in this paper is given on Hardjo titres, it would be best to group together all serovar Hardjo strains in the battery list as well as in Table 1;

NOTE 2- Very important: According to Chiareli et al., Pesq. Vet.Bras. 2012, 32, 238 633-639 and Cosate et al., BMC Vet Res 13, 177 (2017) https://doi.org/10.1186/s12917-017-1081-9, strain Bolivia belongs to genotype Hardjoprajitno and not to genotype Hardjobovis, as it is stated in Table 1. This must be corrected;

Note 3: Hardjobovis is the name of a genotype, not of a serovar. This must be corrected in the Table 1;

  • Correction performed

ines 86-87

Table 1

1)      It is not understood why it is stated in the Table title “Data from Chiareli et al. [16]”? What does it mean and why is this reference needed? To what is it related? This is not clear and, if possible, the reference could be eliminated from the table or else better explained its relation with the table contents. 

2)      The following corrections are suggested:

 Table 1. Standards of Leptospira spp. per serogroup, serovar and strain used in the microscopic agglutination test (MAT) to detect anti-Leptospira antibodies in buffaloes in Marajó Island, Pará state (Brazilian Amazon).

Serogroup

Serovar

Strain/genotype

Bataviae

Australis

Icterohaemorrhagiae

Pomona

Sejroe

Sejroe

Sejroe

Sejroe

Bataviae

Bratislava

Icterohaemorrhagiae

Pomona

Hardjo

Hardjo

Hardjo

Hardjo

Swart

Jez Bratislava

RGA

Pomona

Hardjobovis

CTG/Hardjoprajitno

OMS/Hardjoprajitno

Bolivia/Hardjoprajitno*

* Chiareli et al., Pesq. Vet.Bras. 2012, 32, 238 633-639; Cosate et al., BMC Vet Res 13, 177 (2017);

  • Correction performed
  • *Data entered on lines 42-44 in the introduction. “...none used Leptospira interrogansserovar Hardjo (strain Bolivia, genotype Hardjoprajitno), isolated from cattle in Brazil [9,10]…”

Lines 88-89

Materials and Methods

In:    “The microscopic agglutination test (MAT) was the one of screening, considering positive animals with titration (wrong, titration is the procedure to find the titre, not the same than titre) of 1:100.”

Consider: In the previous page it was already mentioned that the MAT was used (this is a repetition), but so be it. However, correct as follows: “The microscopic agglutination test (MAT) was the one used of for screening and animals with titration titres of 1:100 were considering positive..”

Authors must also make clear if they only screened the sera at this dilution (1:100) ONLY or if they titrated the sera to find the final titre if sera had a titre > 1:100.

  • Correction performed

Lines 89-92

Materials and Methods

The description authors give for the interpretation of MAT results: “We considered animals with zero result or with one cross as non-reactive to the tested serovars, and animals with two to four crosses as sero-reagent…” is not a correct/ideal way to describe the interpretation of the agglutination. Authors are advised to use the usual descriptions widely used internationally and, that is ++ corresponds to 50% agglutination, thus, should re-write the sentence to something like:

“We considered animals as reactive (positive) to the tested serovars if at least 50% agglutination was observed in the respective sera at the 1:100 screening dilution, and non-reactive if less than 50% or no agglutination was observed.”      

  • Correction performed

Line 98

Results

Correct: …(Brazilian Amazon) using the microscopic agglutination test (MAT).

  • Correction performed

Line 100-101

Results

Correct: “…Hardjo (OMS sample strain) and Hardjo type hardjo-bovis Hardjobovis serovars (Table 3). Other serovars 100 that reacted in part of the animals analyzed were the Pomona and Hardjo (CTG sample strain).”

  • Correction performed

Line 106

Results

Correct: …(Brazilian Amazon) using serum agglutination microscopy (SAM) the microscopic agglutination test (MAT).

  • Correction performed

ines 106-107

Table 3

Correct: Where it is written “sample” (CGT and OMS sample) it should be “strain”;

Where is it written “hardjo-bovis” should be “Hardjobovis”

  • Correction performed

Line 108

Discussion

Correct: “…of reactive buffalos than that the one reported in other Brazilian…”

  • Correction performed

Line 111

Discussion

Instead of:    “…studied, that evidence the environmental characteristics…”

Write: “…studied evidence the environmental characteristics…”

  • Correction performed

Lines 114-115

Discussion

Correct: “…of the bacteria Leptospira spp..”

  • Correction performed

Lines 119-120

Discussion

Correct: “…to adopt un hygienic-sanitary management practices,..”

  • Correction performed

Line 123

Discussion

Instead of:    “…with wild animals, whit rodents, from adjacent forest areas,..”

Write: “…with wild animals, whit namely rodents, from adjacent forest areas,..”

Or, better still: “…with wild animals, namely rodents and other small mammals, from adjacent forest areas...

  • Correction performed

Line 129

Discussion

Correct: “…between productive production and wild animals,…”

  • Correction performed

Lines 133-134

Discussion

“This is the first study to report the infection of serovar Hardjo (Bolivia)* in buffaloes in Marajó Island, Amazon Biome, Brazil.”

* Note: You cannot state this scientifically because Serology is only serogroup indicative. To evaluate the presence of a given serovar/let alone strain (that is the infection), it is necessary to isolate, identify and characterise the strain involved. To conclude that the buffaloes are infected with Bolivia, you had to isolate the strain.

So, rephrase: “This is the first study to report leptospiral titres to serovar Hardjo (Bolivia) in buffaloes in Marajó Island, Amazon Biome, Brazil.”

  • Correction performed

Line 141

Discussion

Correct: “…, as well as the isolation and identification of the bacterial agents, (remove comma) in buffaloes…”

  • Correction performed

Line 150

Discussion

Correct: “…to be adopted towards the herds.”

Line 155-156

Discussion

Correct: “…with horses, one of the hosts and source of infection by this specific serovar.”

Note: there are other hosts for Bratislava (namely pigs and hedgehogs)

Line 157-158

Discussion

Correct: “…were the least common titres found in the present study among our samples, which…”

  • Correction performed

Lines 163-167

Discussion

Correct: “The occurrence of Icterohaemorrhagiae titres in ruminants (in the buffaloes?) was low, it was still, however, as well as other serovars reactive in the present study, they still reflect are risk factors for the zoonotic occurrence of the disease and its detection it is reiterates the importance of the use of different serovars in the antigen battery of diagnostic tests of leptospirosis.

Lines 170-171

Conclusion

Correct: “The titres against serovar Hardjo (strain Bolivia) were the most prevalent.”

Lines 174-175

Conclusion

Correct: “..It is important to carry out serological and bacteriological surveys in order to identify the serovars that occurin the herds...”

  • Correction performed

Reviewer 2 Report

Line 62: Mention the geographical description of the sampling area, describe how the selection of the animals was, as well as the sampling design.

Improve discussion and conclusion.

Author Response

Reviewer 2

We appreciate all aspects of reviewer 3's comments and suggestions, which made it possible to make this manuscript valid for all readers. Thank you very much for the compliments and empathy in the evaluation.

  • Line 62: Mention the geographical description of the sampling area, describe how the selection of the animals was, as well as the sampling design. Improve discussion and conclusion. Dear reviewer, the samples used were obtained from a serum bank, from animals sent for slaughter, in the nine municipalities of Ilha de Marajó. A specific sample design was not made, but the use of all available sera that were representative of the region. As the animals are slaughtered in municipal inspection slaughterhouses, we do not have the traceability of the data, we only know, by the Animal Transit Guide, required for the slaughter of the animals, that they were on average 36 months old (approximately 3 years), for both the sexes and which municipality they were from. These samples were collected by the veterinarians present at the slaughter line and sent to the Institute of Veterinary Medicine of the Fderal University of Pará and stored in a bank of biological samples that we have at the Veterinary Hospital. When we reached a minimum statistical number to carry out the study, we proceeded to carry out the MAT. So much so that the discussion is about the Island of Marajó and not about the municipalities in isolation, as we had, for example, the municipality of Anajás with only 8 samples and the municipality of Soure with 60 samples. Sorry for not being able to address this point, but we have improved the article in accordance with your requests and those of the other two reviewers.

Best regards,

Felipe Masiero Salvarani

Reviewer 3 Report

The manuscript entitled “Anti-Leptospira antibodies in buffaloes in the Marajó Island“ aims to determine the occurrence of antibodies against Leptospira spp., including serovar Hardjo (Bolivia) isolated from cattle in Brazil and underused in buffaloes. Authors also reiterated the importance of the use of different serovars in the battery of diagnostic tests of leptospirosis in livestock.

There is a good idea to determine the occurrence of 16 antibodies against Leptospira spp., including serovar Hardjo (Bolivia), in buffaloes.
The following are details that I’d like to comment and provide suggestions.

I hope that all aspects of the comments and suggestions would be helpful to make this manuscript sound for all readers.

The following are details that I’d like to comment and provide suggestions.

Abstract:

·       The conclusion should be added some more specific rational of the manuscript to support the importance of the use of different serovars in the battery of diagnostic tests of leptospirosis in livestock.  

Introduction:

            -

Methods:

            -

Results:

·       Page 3; Table 2, It would be more informative to provide number and percentage of buffaloes with seropositive up to two serovars for each municipality.

Discussion:

·       Page 3-4, Lines 112 – 114; Authors discussed that the endemicity factors for leptospirosis, such as; levels pf precipitation, relative humidity, average annual temperature, could facilitated the high incident of seropositive of buffaloes in Marajó Island. I’d recommend authors to do more comparison factors with really exist data in São Paulo, Paraíba, Maranhão. So, you may have more clear data to discuss with all these factors.

·       Page 4, Lines 117 – 124 and Lines 151 – 156; Authors should provide comparison data from validated or government agencies at the Marajó Island (if possible, at the same period as buffaloes’ samples were collected) for the prevalence of seropositive in the stated animal such as; rodents, rat, pig, human, as well as wild animal (if data available). The discussion would be more sound and more informative.

Conclusion and Abstract should be minor modified as suggestions and comments above.

 I hope that all aspects of the comments and suggestions would be helpful to make this manuscript sound for all readers.

Author Response

Dear Reviewer 3

We appreciate all aspects of reviewer 3's comments and suggestions, which made it possible to make this manuscript valid for all readers. Thank you very much for the compliments and empathy in the evaluation.

  • Conclusion and Abstract were modified as suggestions and comments lines 22 to 29 (abstract) and lines 178 and 180 (conclusion)
  • Results modified and inserted in the text, but not in the table, as we believe that the table would not be as didactic. Then insert in rows 111 to 112 the number and percentage of buffalo seropositive for more than two serovars in the MAT, which was 263 animals or 67.2% of animals. ”Of all buffalo tested 263 (67.2%) were seropositive up to two serovars in this research.”
  • Discussion: Page 3-4, Lines 112 – 114; Authors discussed that the endemicity factors for leptospirosis, such as; levels pf precipitation, relative humidity, average annual temperature, could facilitated the high incident of seropositive of buffaloes in Marajó Island. I'd recommend authors to do more comparison factors with really exist data in São Paulo, Paraíba, Maranhão. So, you may have more clear data to discuss with all these factors. We do not find it necessary to describe the data from Ilha de Marajó, as it is in the public domain, and therefore from the international community, which has very attentive eyes for the Amazon Biome, that actually the levels of precipitation, relative humidity, average annual temperature, could have facilitated the high incidence of seropositive buffaloes on Ilha de Marajó, as they are totally different from those found in articles that report these same endemicity factors in the states of São Paulo, Paraíba, Maranhão and even in the "mainland region" of the state of Pará. As Marajó is a fluvial-maritime island, directly influenced by climatic conditions and "water dictatorships" with large extensions of flooded areas used for animal breeding, especially buffaloes, we do believe that the specificity of the endemicity factors on Marajó Island is a of the main explanations for more than 90% of seropositive animals for leptospirosis. And unfortunately, the articles used in this discussion do not specifically mention the average values of precipitation levels, relative humidity, average annual temperature, so we cannot specify more, but rather suggest that these endemicity factors (climate) are possible factors of greater predisposition to occurrence of leptospirosis on the Island of Marajó, with annual rainfall greater than 4000mm (Lima et al., 2005. The Marajó Island: historical revision, hydroclimatology, hydrographical basins and management proposals raphics and management proposals. Holos Environment, 5:(1), p.65-80.) and (Reis et al., 2005. Characterization of waters in the region of Marajó through concentrations of 0-18 and D. Acta Amazonica, 7:(2), p-.209-222.)
  • Discussion: Page 4, Lines 117 – 124 and Lines 151 – 156; Authors should provide comparison data from validated or government agencies at the Marajó Island (if possible, at the same period as buffaloes’ samples were collected) for the prevalence of seropositive in the animal stated such as; rodents, rat, pig, human, as well as wild animal (if data available). The discussion would be more sound and more informative. We understand the suggestion however there are no data from validated or government agencies at the Marajó Island in the literature. But we believe that the discussion is solid and informative, it could be more, but unfortunately there are no official data to be referenced.

Best regards,

Felipe Masiero Salvarani

Round 2

Reviewer 3 Report

Line 192-196: Rewrite the sentence ..."The occurrence of Icterohaemorrhagiae titres in ruminants (in particular in buffaloes) was low, it was still, however, as well as other serovars reactive in the present study, they still reflect are risk factors for the zoonotic occurrence of the disease and its detection it is reiterates the importance of the use of different serovars in the antigen battery of diagnostic tests of leptospirosis..." for more understandable and grammartical correction.

Line 192-196: Rewrite the sentence ..."The occurrence of Icterohaemorrhagiae titres in ruminants (in particular in buffaloes) was low, it was still, however, as well as other serovars reactive in the present study, they still reflect are risk factors for the zoonotic occurrence of the disease and its detection it is reiterates the importance of the use of different serovars in the antigen battery of diagnostic tests of leptospirosis..." for more understandable and grammatical correction.

Author Response

Reviewer 3
Thanks for the new suggestion on rewrite the sentence for more understandable and grammartical correction. I would like to emphasize that this change made in Lines 192-196: "The occurrence of Icterohaemorrhagiae titres in ruminants (in particular in buffaloes) was low, it was still, however, as well as other serovars reactive in the present study, they still reflect are risk factors for the zoonotic occurrence of the disease and its detection it is reiterates the importance of the use of different serovars in the antigen battery of diagnostic tests of leptospirosis" was a requirement of reviewer number 1 and was placed in the text in accordance with his indication. But we agreed to rewrite the sentence which was like this and I hope it agrees with what you expect: " The occurrence of animals with antibodies titers against serovar Icterohaemorrhagiae was lower, as well as for other serovars (Bataviae and Bratislava) in the present study and in literature. However, all serovars of Leptsopira are risk factors for the zoonotic occurrence of the disease, reiterating the importance of in use of maximun of serovars in the battery of diagnostic tests for leptospirosis in animals and humans."

Thanks again for the opportunity to improve our work with your comments.

Best regards,

Felipe Masiero Salvarani